# Magnetofluorescent Nanocomposite Comprised of Carboxymethyl Dextran Coated Superparamagnetic Iron Oxide Nanoparticles and β-Diketon Coordinated Europium Complexes

**DOI:** 10.3390/nano9010062

**Published:** 2019-01-04

**Authors:** Daewon Han, Seung-Yun Han, Nam Seob Lee, Jongdae Shin, Young Gil Jeong, Hwan-Woo Park, Do Kyung Kim

**Affiliations:** 1Department of Cell Biology, Konyang University College of Medicine, Daejeon 302-718, Korea; eodnjs0319@naver.com (D.H.); shinjd@konyang.ac.kr (J.S.); 2Department of Anatomy, College of Medicine, Konyang University, Daejeon 302-718, Korea; jjzzy@konyang.ac.kr (S.-Y.H.); nslee@konyang.ac.kr (N.S.L.); ygjeong@konyang.ac.kr (Y.G.J.); 3Myunggok Medical Research Institute, Konyang University College of Medicine, Daejeon 302-718, Korea

**Keywords:** europium complexes, superparamagnetic, iron oxide nanoparticles, magnetofluorescent

## Abstract

Red emitting europium (III) complexes Eu(TFAAN)_3_(P(Oct)_3_)_3_ (TFAAN = 2-(4,4,4-Trifluoroacetoacetyl)naphthalene, P(Oct)_3_ = trioctylphosphine) chelated on carboxymethyl dextran coated superparamagnetic iron oxide nanoparticles (CMD-SPIONs) was synthesized and the step wise synthetic process was reported. All the excitation spectra of distinctive photoluminesces were originated from f-f transition of Eu^III^ with a strong red emission. The emission peaks are due to the hypersensitive transition ^5^D_0_→^7^F_2_ at 621 nm and ^5^D_0_→^7^F_1_ at 597 nm, ^5^D_0_→^7^F_0_ at 584 nm. No significant change in PL properties due to addition of CMD-SPIONs was observed. The cytotoxic effects of different concentrations and incubation times of Eu(TFAAN)_3_(P(Oct)_3_)_3_ chelated CMD-SPIONs were evaluated in HEK293T and HepG2 cells using the WST assay. The results imply that Eu(TFAAN)_3_(P(Oct)_3_)_3_ chelated CMD-SPIONs are not affecting the cell viability without altering the apoptosis and necrosis in the range of 10 to 240 μg/mL concentrations.

## 1. Introduction

Non-invasive fluorescent [1] probes are required to evaluate the efficacy of the drug at the molecular level. The evaluation of the efficacy of the drug is also important from a biological point of view, as it traces the specific expression after administrating the drug into the cells [2]. Among these probes, fluorescence probes are excellent for tracking light emitted from the molecules after excitation at a specific wavelength because they have high sensitivity and spatial resolution [3]. One of the most important factor in selecting a bioimaging probe is the organic molecules/particles needed to be easily localize/internalize to particular organelles or sub-cellular sites. Currently, optical probes are classified into two categories; one is organic molecules such as fluorescein isothiocyanate (FITC) [4], Alexa Fluor [5], tetramethylrhodamine (TRITC) [6], cyanine (Cy3, Cy5, and Cy7), green fluorescent protein (GFP), yellow fluorescent protein (YFP), and cyan fluorescent protein (CFP), etc., while the other consists of inorganic materials/quantum dots (QDs), such as CdSe/ZnS [7], InP/ZnS [8], CuInS/ZnS [9], and CH_3_NH_3_PbX_3_ (X = Cl, Br, I) [10], etc.

QDs are very commonly used materials in biomedical applications such as in detection, biomarkers, and imaging agents. However, the ineffective uptake of QDs in living cells is an obstacle to its use in nanomedicine. A high dose of QDs is required to resolve the low localization in living cells. It is therefore significant to realize and strategy of designing an effective QDs for biomedical uses by understanding the permeability of cell membranes depending on the substances and functional groups present on the QD surface [11].

The synthesis of lanthanide complexes, which is a substance that emits light under ultraviolet light, has been extensively studied because of its broad range of applications. In recent years, the lanthanide complexes have been intensively investigated in nanomedicine because of their strong characteristics against anti-photobleaching compared with conventional fluorescent materials. They can be used in diagnostic kit, micro chemical detections, cell labeling, and other applications, because of their low cytotoxicity and high quantum yield [12]. 

At present, magnetic nanoparticles have been developed to improve the diagnosis and therapeutic effects of various diseases, and they are the most studied materials among nano-sized materials developed to date. Superparamagnetic iron oxide nanoparticles (SPION) have inherent beneficial characteristics, including magnetic properties, magnetized under external magnetic field, biocompatibility, high dispersivity after coating with biocompatible substance, cellular uptake, and a short blood half-life. These benefits are the most effective parameters for transporting drugs, proteins, and probes for nanomedicinal applications [13,14].

In this work, we try to develop a multi-purpose nanoprobe called magnetofluorescent [15,16] nanoparticles that can be traced noninvasively by MRI in-vivo and by confocal microscope in-vitro. SPIONs were prepared in the presence of dextran molecules and carboxymethyl group was activated with monochloroacetic acid followed by crosslinking the dextran (CLD) molecules with epichlorohydrin to cage-like structure tightly grafted on SPIONs. Finally, β-diketon coordinated europium complexes composed of 2-(4,4,4-trifluoroacetoacetyl)naphthalene (TFAAN), trioctylphophine (P(Oct)_3_), and Eu^III^ were chelated on CLD-SPION.

## 2. Materials and Methods

### 2.1. Chemicals

Ferric chloride hexahydrate (FeCl_3_·6H_2_O), ferrous chloride tetrahydrate (FeCl_2_·4H_2_O), ammonium hydroxide solution (28% in water), europium chloride hexahydrate (EuCl_3_·6H_2_O), trioctylphophine (P(Oct)_3_), 3-[4,5-dimethylthiazol-2-yl]-2,5-diphenyltetrazolium bromide (MTT), and ethanol dialysis tubing cellulose membrane with a weight-cutoff (MWCO = 12,400) were purchased from Sigma-Aldrich Chemical Co. (St. Louis, MO, USA). Tokyo Chemical Industry Co. Ltd (Tokyo, Japan) supplied the 2-(4,4,4-Trifluoroacetoacetyl)naphthalene (TFAAN). Dextran T-10, epichlorohydrin and monochloroacetic acid (MCA) were supplied by DaeJung Chemicals and Metals Co. Ltd. (Kyunggi do, South Korea).

### 2.2. Characterization

The particle size, morphology, and distribution quality of Eu(TFAAN)_3_(P(Oct)_3_)_3_ on CMD-SPION nanospheres were analyzed by Hitachi H-7650 TEM (EVISA, Tokyo, Japan). Fourier transform infrared (FT-IR) spectra were recorded using an ALPHA FT-IR Spectrometer equipped with Platinum ATR (Bruker, Billerica, MA, USA). Excitation and emission spectra were examined by fluorospectrophotometer (RF-5301PC Shimadzu, Kyoto, Japan) equipped with a 150 W xenon discharge lamp as a light source. Cells were observed using LSM 510 confocal laser scan microscope (Zessi, Inc., San Diego, CA, USA) and E800 epifluorescence microscopic (Nikon, Inc., Tokyo, Japan). 

### 2.3. Synthesis of β-Diketon Coordinated Europium Complexes

Europium complexes [17,18], Eu(TFAAN)_3_(P(Oct)_3_)_3_, was prepared based on previously reported method with a minor modification. Europium chloride hexahydrate (EuCl_3_·6H_2_O, 0.366 g, 1 mmol) was dissolved in 50 mL distilled H_2_O. In similar way, 2-(4,4,4-Trifluoroacetoacetyl)naphthalene (TFAAN, 0.266 g, 1 mmol) and trioctylphosphine (P(Oct)_3_, 0.370 g, 1 mmol) were dissolved in 50 mL EtOH. The final concentration of each stock solutions were 0.02 mmol. From the stock solution, 2 mL Eu^3+^, 6 mL TFAAN, 6 mL P(Oct)_3_, and 100 μL (28% in water) NH_4_OH were transferred to 20 mL glass bottle with a capped tightly and the temperature was increased to 60 °C in a water bath under magnetic stirring. The product was precipitated by centrifuge, washed three times with distilled H_2_O and dried in vacuum oven at 70 °C for 24 h. The chemical structure of Eu(TFAAN)_3_(P(Oct)_3_)_3_ is shown in Scheme 1a. 

### 2.4. In situ Synthesis of Dextran Coated Superparamagnetic Iron Oxide Nanoparticles by Co-Precipitation

Scheme 1b shows the overall procedure to prepare the CMD-SPIONs. N_2_ gas was directly flowed into the solution for 30 min to remove the oxygen before the experiments. 5.46 g Iron(III) chloride hexahydrate and 1.99 g iron(II) chloride tetrahydrate were dissolved in 90 mL H_2_O and 500 μL concentrated HCl by heating at 60 °C in a water bath until all the salt was fully dissolved. After a transparent solution was achieved, the solution was topped up with H_2_O to make a final volume of 100 mL to prepare the iron stock solution ([Fe^2+^] = 200 mM, [Fe^3+^] = 100 mM). After this, 1 g Dextran T-10 (Mw 10,000) was dissolved in 10 mL iron stock and add 90 mL H_2_O. After the dextran was fully dissolved, the solution was placed in an ice bath for 30 min. 10 mL concentrated ammonia was added dropwise under magnetic stirring and kept for 30 min. The temperature of the mixture was increased to 70 °C and kept for an additional 30 min. After the product was cooled to 25 °C (R. T.), the black solid was recovered by neodymium magnet and the supernatant was eliminated by decantation. The solid precipitate was washed 5 times with H_2_O to remove unreacted salts and by-products. The final product, called SPIONs, was dispersed in 100 mL H_2_O by probe sonicate for 1 min in an ice bath to form an extremely stable colloidal solution. Crosslinking between the dextran molecules surrounding on SPIONs was performed to attach the dextran more steadily by following process; 10 mL 5 M NaOH was added in stock solution of SPIONs under vigorous magnetic stirring. 5 mL epichlorohydrin was added into the mixture and kept for 24 h at 25 °C under vigorous shaking by the shaker to avoid the phase separation between aqueous and organic layer. The constant shaking stimulates the chemical reaction between two different phases. After finalizing the reaction, the mixture solution was poured in dialysis tubing cellulose membrane with a weight-cutoff (MWCO = 12,400) and dialyzed against 5 L distilled H_2_O. The distilled H_2_O was changed with fresh one every 1 h for 5 times and left overnight. The conductivity of the H_2_O was monitored by a conductivity meter to regulate the termination point of the washing progression. The solution was concentrated by vacuum dryer until a final volume of 20 mL and remarked as CLD-SPIONs. To activate the carboxylic group in dextran, carboxymethylation was accomplished though the following process; 10 mL 0.1 M NaOH was mixed with 10 mL CLD-SPIONs under magnetic stirring for 30 min at 25 °C while directly purging N_2_. After this, 0.443 MCA was added dropwise to the solution, and the temperature was increased to 60 °C in an oil bath and kept for 60 min while N_2_ was flowing. After the reaction was completed, CMD-SPIONs was dialyzed against 5 L distilled H_2_O to remove the impurities and kept at 4 °C.

### 2.5. Magnetofluorescent Composite of Eu(TFAAN)_3_(P(Oct)_3_)_3_@CMD-SPIONs

30 mg CMD-SPIONs in 10 mL deionized water was added in 1 mg Eu(TFAAN)_3_(P(Oct)_3_)_3_ in 10 mL EtOH and heated to 80 °C in oil bath under magnetic stirring for 1 h. The sample was dialyzed against 5 L deionized water using a membrane tubing with a molecular cut-off 12,500 for three consecutive periods of 8 h. The final chemical structure of Eu(TFAAN)_3_(P(Oct)_3_)_3_@CMD-SPIONs is shown in Scheme 1c. 

### 2.6. Cell Culture

Human embryonic kidney (HEK) 293T cells and human liver cancer cell line HepG2 cells were cultured in Dulbecco’s Modified Eagle’s Medium (DMEM, Welgene, Gyeongsangbuk-do, Korea) supplemented with 10% fetal bovine serum (FBS, Welgene, Korea) and 100 U/mL penicillin-streptomycin (Welgene, Korea). Culture conditions were maintained in a humidified atmosphere containing 5% CO_2_ at 37 °C. 

### 2.7. Immunofluorescence

HEK293T cells or HepG2 cells were seeded onto each coverslip with 3 × 10^5^ cells per coverslip in 24-well culture plates and then grown to 80% confluence. The cells were treated with indicated concentrations of Eu(TFAAN)_3_(P(Oct)_3_)_3_@CMD-SPION_S_ dissolved in water for indicated time periods. Coverslips were washed once with phosphate-buffered saline (PBS) and fixed with 4% paraformaldehyde. After washing, coverslips were mounted in 4′,6-diamidino-2-phenylindole (DAPI, Invitrogen, Carlsbad, CA, USA) on glass slides. Samples were analyzed under an epifluorescence-equipped microscope (DM2500, Leica, Wetzlar, Germay).

### 2.8. Cytotoxicity Assay

Cell viability was measured using WST-1 assay (Daeil Lab Service) according to the standard protocol of the manufacturer. Briefly, HEK293T cells and HepG2 cells were plated in 96-well plates at a concentration of 1 × 10^4^ cells/well and treated with indicated concentrations of Eu(TFAAN)_3_(P(Oct)_3_)_3_@CMD-SPIONS for indicated time periods. After incubated, 10 µL of WST-1 solution was added to each well and incubated for 30 min at 37 °C under 5% CO_2_ incubator. After this, optical density of 96-well plates was measured in a microplate reader (Bio-Rad) at 450 nm and the absorbance values of the treated cells were expressed as a percentage of the absorbance values of the control. 

## 3. Results

Figure 1 shows TEM images of carboxymethyl dextran coated superparamagnetic iron oxide nanoparticles (CMD-SPIONs) and Eu(TFAAN)_3_(P(Oct)_3_)_3_ conjugated on CMD-SPIONs. The average particle diameter of SPIONs was around 12 nm with an irregular shape. After conjugation of Eu(TFAAN)_3_(P(Oct)_3_)_3_, the particles had some agglomeration. The gray molecules are Eu(TFAAN)_3_(P(Oct)_3_)_3_, as shown in Figure 1b and can be eliminated by washing with ethanol by centrifuge or applying an external magnetic forces such as neodymium magnet.

Differential scanning calorimetry (DSC) analysis was performed for CMD-SPION, Eu(TFAAN)_3_(P(Oct)_3_)_3_, and Eu(TFAAN)_3_(P(Oct)_3_)_3_ chelated CMD-SPIONs (Figure 2). DSC plot of CMD-SPION exhibited the endothermic peaks at 75 °C and 164.6 °C and exothermic peak at 136 °C, whereas Eu(TFAAN)_3_(P(Oct)_3_)_3_ and Eu(TFAAN)_3_(P(Oct)_3_)_3_ chelated CMD SPIONs did not appear at all. The temperature profile of Eu(TFAAN)_3_(P(Oct)_3_)_3_ system showed the distinctive peak at 54 °C as it appeared in DSC curve of both Eu(TFAAN)_3_(P(Oct)_3_)_3_ and Eu(TFAAN)_3_(P(Oct)_3_)_3_ chelated CMD SPIONs.

Figure 3 shows PL spectra of (a) emission and (b) excitation profiles depending on different concentration of Eu(TFAAN)_3_(P(Oct)_3_)_3_ chelated on CMD-SPIONs. All the excitation spectra reveal a comparable tendency of distinctive photoluminesces (PL) coming from f-f transition of Eu^III^ with a strong red emission [19]. The maximum emission (EM) peak is due to the hypersensitive transition ^5^D_0_→^7^F_2_ at 621 nm and ^5^D_0_→^7^F_1_ at 597 nm, ^5^D_0_→^7^F_0_ at 584 nm. No significant change in PL properties due to addition of CMD-SPIONs was observed.

Figure 4 shows zeta potentials of CMD-SPIONs and Eu(TFAAN)_3_(P(Oct)_3_)_3_ chelated on CMD-SPIONs. Zeta (*ζ*) potential is used as a proper index for colloidal stability by evaluating the quantification of the magnitude of the surface charge on the particles [20], and the value is generally used to interpret and manipulate the stability of colloidal dispersion. The *ζ*-potentials of CMD-SPIONs is −17 mV and Eu(TFAAN)_3_(P(Oct)_3_)_3_ chelated on CMD-SPIONs is changed to 9.7 mV.

Figure 5 shows FTIR spectra of Eu(TFAAN)_3_(P(Oct)_3_)_3_, CMD-SPIONs and Eu(TFAAN)_3_(P(Oct)_3_)_3_ chelated on CMD-SPIONs show absorption peaks around 3300 cm^−1^ coming from OH-stretching vibrations of water molecules and 2922 cm^−1^ can be allocated to the sp^3^ bonding of C–H. In the CMD spectrum, deformation vibration δ(C–OH) which appears at 1250 cm^−1^ and ν(C–O) vibration around 1150 cm^−1^. The absorption at 1462 cm^−1^ is attributed to C=C bond and 1688 cm^−1^ is assigned to C=O bond. The stretching vibration bands of P=O (1061 cm^−1^) bond are appeared in Eu(TFAAN)_3_(P(Oct)_3_)_3_ and Eu(TFAAN)_3_(P(Oct)_3_)_3_ chelated on CMD-SPIONs. These results imply that Eu(TFAAN)_3_(P(Oct)_3_)_3_ was successively grafted on magnetic nanoparticles. 

The cytotoxic effects of concentrations and incubation times of Eu(TFAAN)_3_(P(Oct)_3_)_3_ chelated CMD-SPIONs were evaluated in HEK293T and HepG2 cells using the WST assay. (Figure 6) Figure 6a shows the cytotoxic effects of HEK293T and HepG2 cells treated with the indicated concentration of Eu(TFAAN)_3_(P(Oct)_3_)_3_@CMD-SPIONs for 6 h. Cell viability was measured by WST-1 assay. Data are shown as mean ± s.e.m. (*n* = 3). Figure 6b shows cytotoxic effects of HEK293T and HepG2 cells treated with 72 μg/mL Eu(TFAAN)_3_(P(Oct)_3_)_3_@CMD-SPIONs for indicated periods of time. Cell viability was measured by WST-1 assay. Data are shown as mean ± s.e.m. (*n* = 3) Figure 6c,d shows the results of HEK293T (c) and HepG2 cells (d) treated the indicated concentration of Eu(TFAAN)_3_(P(Oct)_3_)_3_@CMD-SPIONs for 24 h. The cells were stained with Muse Annexin V and Dead Cell reagent and then analyzed for apoptosis by the Muse Cell Analyzer.

Figure 7a,b shows intracellular uptake and distribution of Eu(TFAAN)_3_(P(Oct)_3_)_3_@CMD-SPIONs in HEK293T and HepG2 cells. Representative images of HEK293T (a) and HepG2 cells (b) treated with vehicle (Control) or Eu(TFAAN)_3_(P(Oct)_3_)_3_@CMD-SPIONs at 72 μg/mL concentration for 4 h. Nuclei were stained with DAPI (blue). Scale bar, 20 µm.

## 4. Discussion

The crosslinking reaction of dextran [21] by epichlorohydrin happen in inter- or intra-molecular forms and are grafted on SPIONs surfaces. The dextran molecules are transformed from the surface of SPIONs into a strong and rigid structure, resulting in a heterogeneous solid structure on the SPIONs surface. The core-shell structure maintains the intermolecular bonding of physically bonded intermolecular materials and prevents separation in the aqueous solution. The epoxy moiety of epichlorohydrin alkylates OH groups and its epoxy group interacts with other OH groups of the dextran to form the corresponding inter- and intramolecular crosslinks. 

In general, the magnetic nanoparticles have remanence magnetization even though it is classified as superparamagnetic material. The residual magnetic forces will induce the interactions between the magnetic particles resulting in large coagulation in the diameter range of 50–300 nm. To avoid the agglomeration, nonmagnetic substances were grafted on the surface of SPIONs. In this study, in situ formation of SPIONs was introduced to resolve the difficulties caused by coagulation during the synthesis of magnetic nanoparticles. Dextran molecules in the water form polymeric matrix and Fe^2+^ and Fe^3+^ ions are located in dextran molecules. In this manner, SPIONs nucleate inside the matrix and retain the distance among the particles while the particles are grown by thermal energy. The inter-particular forces due to the dipole-dipole interaction are adequate to elude the stimulus of residual magnetization resulting in forming exceptionally constant colloidal suspension. More benefits of using carbohydrate (i.e., dextran) are that it is biocompatible along with versatile derivatives by activation of specific functional groups. The DSC peak at 54 °C was disappeared because TFAAN and P(Oct)_3_ molecules were diffused into each other resulting in disappearing the endothermic and exothermic peaks, indicating that each molecule was changed into amorphous phase. In addition, the mass of Eu(TFAAN)_3_(P(Oct)_3_)_3_ is relatively larger than that of CMD-SPIONs when they are grafted and is not observed even with exothermic or endothermic peaks.

Approximately 20% of EM quenching was observed by comparing the EM intensity of Eu(TFAAN)_3_(P(Oct)_3_)_3_ chelated on CMD-SPIONs and Eu(TFAAN)_3_(P(Oct)_3_)_3_ at 621 nm. As the concentration of Eu(TFAAN)_3_(P(Oct)_3_)_3_ against CMD-SPIONs was increased, ^5^D_0_→^7^F_2_ was proportionally amplified with a virtually Gaussian shape. 

The PL of Eu(TFAAN)_3_(P(Oct)_3_)_3_ and Eu(TFAAN)_3_(P(Oct)_3_)_3_@CMD-SPIONS is analyzed by PL intensity ratio of ^5^D_0_→^7^F_2_ and ^5^D_0_→^7^F_1_. ^5^D_0_→^7^F_1_ is comparatively strong [22], which corresponds to the magnetic dipole transition, has an independent intensity value inherently not affected by coordination environment. Therefore, the intensity values of the ^5^D_0_→^7^F_1_ transition that are forbidden both for magnetic and electric dipole and can be used for comparison. In opposition, the intensity value of ^5^D_0_→^7^F_2_ is an electric dipole transition affected by the physicochemical change values around the Eu^III^. 

*ζ*-potentials of CMD-SPIONs have a rather lower value but are extremely stable in water because CMD on SPIONs forms hyper branched matrix in water resulting in localization of SPIONs inside the net-like matrix. For this reason, CMD-SPIONs are exceptionally consistent without coagulation/flocculation even though the value of *ζ*-potentials is relatively low. The net electro charge on CMD-SPION shows opposite values after modifying the surface with Eu(TFAAN)_3_(P(Oct)_3_)_3_. Figure 4b–d shows digital images of CMD-SPION (left) and Eu(TFAAN)_3_(P(Oct)_3_)_3_ chelated on CMD-SPIONs (right) dispersed in water under (b) daylight, (c) daylight + UV light at 365 nm, and (d) UV light at 365 nm. Eu(TFAAN)_3_(P(Oct)_3_)_3_ chelated on CMD-SPIONs exhibited quite bright luminesce with a red color under excitation with a UV light at 365 nm [23].

The cytotoxicity of 6 h after treatment with a different concentration of Eu(TFAAN)_3_(P(Oct)_3_)_3_ chelated CMD-SPIONs (10 to 240 μg/mL) show that the cell viability in the range of 10 to 240 μg/mL concentrations was more than 95%. Eu(TFAAN)_3_(P(Oct)_3_)_3_ chelated CMD-SPIONs were not toxic to HEK293T and HepG2 cells even at a concentration of 240 μg/mL. At all time periods, no significant difference in cell viability was observed for all two cell types above when treated with Eu(TFAAN)_3_(P(Oct)_3_)_3_ chelated CMD-SPIONs (Figure 6b). In addition, we determined the apoptotic effect of Eu(TFAAN)_3_(P(Oct)_3_)_3_ chelated CMD-SPIONs in HEK293T and HepG2 cells by using Annexin V and dead cell reagent labeling flow cytometry. The four-quadrant plots in each panel show the necrotic cells (upper left), the late apoptotic cells (upper right), the viable cells (lower left), and the early apoptotic cells (lower right). Both cells treated with Eu(TFAAN)_3_(P(Oct)_3_)_3_ chelated CMD-SPIONs revealed four-quadrant plots similar to those of the vehicle-treated cells (Figure 6c,d). These results indicate that Eu(TFAAN)_3_(P(Oct)_3_)_3_ chelated CMD-SPIONs do not affect cell viability and do not alter the cell death program [24]. 

The intracellular uptake of Eu(TFAAN)_3_(P(Oct)_3_)_3_ chelated CMD-SPIONs was evaluated in HEK293T and HepG2 cells using an epifluorescence-equipped microscopy. Internalization of Eu(TFAAN)_3_(P(Oct)_3_)_3_ chelated CMD-SPIONs by HEK293T and HepG2 cells was not observed after 1 h incubation. After 6 h incubation, the cellular uptake of Eu(TFAAN)_3_(P(Oct)_3_)_3_ chelated CMD-SPIONs in both cells occurred and was distributed within the cytoplasm (Figure 7). Since the europium complexes synthesized in this study have emission spectra at 619 nm when excited in the UV range, Eu(TFAAN)_3_(P(Oct)_3_)_3_ chelated CMD-SPIONs do not require an additional red tracker.

## 5. Conclusions

We designed and synthesized a novel magnetofluorescent nanoprobe comprising of red emitting europium (III) complexes Eu(TFAAN)_3_(P(Oct)_3_)_3_ chelated on carboxymethyl dextran coated CMD-SPIONs that can be traced noninvasively by MRI in-vivo and by confocal microscope in-vitro. All the excitation spectra distinctive photoluminesces came from f-f transition of Eu^III^ with a strong red emission. No significant change in PL properties due to addition of CMD-SPIONs was observed. The cell viability measured in HEK293T and HepG2 cells shows that Eu(TFAAN)_3_(P(Oct)_3_)_3_ chelated CMD-SPIONs do not affect cell viability and do not alter apoptosis and necrosis in the range of 10 to 240 μg/mL concentrations. At 4 h after incubation, the red fluorescence from Eu(TFAAN)_3_(P(Oct)_3_)_3_ chelated CMD-SPIONs was mainly located in the cytoplasm with no significant cytotoxicity.

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
