# Peer review of "Magnetofluorescent Nanocomposite Comprised of Carboxymethyl Dextran Coated Superparamagnetic Iron Oxide Nanoparticles and β-Diketon Coordinated Europium Complexes"

_nanomaterials, 2019, doi:10.3390/nano9010062_

Reviewer 1 Report

Scheme 1: the last structure of the nanoparticle is not clear so it might be simplified for the readers to understand the surface chemistry of the nanoparticle.

Why Figure 6 and 7 are in the discussion section?

The authors should do TGA analysis to estimate the content of each compounds of the nanoparticle

4. The authors should clearly explain the benefit of dual imaging using MRI and fluorescence. It is not clear that there is a need for bioimaging.

Author Response

1. Scheme 1: the last structure of the nanoparticle is not clear so it might be simplified for the readers to understand the surface chemistry of the nanoparticle.

Response 1: Thanks for the comments. We changed the scheme 1 into the simplified form.

2. Why Figure 6 and 7 are in the discussion section?

Response 2: Thanks for the comments. The Figure 6, Figure 7, and some description were moved to “results” section. 

3. The authors should do TGA analysis to estimate the content of each compounds of the nanoparticle

Response 3: Thanks for the comments. Unfortunately, we don’t have TGA in our institute. If we want to include the TGA data, it will take at least one month because it is end of this year and beginning of new year. It is difficult to arrange instrumentation by other institute. Due to the time limitation of reviewing process (3days), we could not insert it in this stage.

4. The authors should clearly explain the benefit of dual imaging using MRI and fluorescence. It is not clear that there is a need for bioimaging.

Response 4: For example, a specific antibody was conjugated on suggested magnetofluorescent probe, the target legion such as cancer and area of interesting (AOI) can be straightforwardly visualized by MRI. However, it is quite ambiguous to identify the target legion that should be removed during the surgical operation. If we use magnetofluorescent probe, the legion could be easily identified simply by irradiating UV light during the operation.  

Reviewer 2 Report

In this manuscript by Han and Co-authors presented their results on the newly synthesized dextran coated SPOINS with carboxymethyl group, chelated with europium complexes.   These newly synthesized SPOINS were thoroughly characterized and evaluated for their biocompatibility using the HEK293T and HepG2 cells.  Biocompatibility studies included multiple assays which includes cytotoxicity by WST and apoptosis using Annexin V.   Overall the manuscript adds to the biomedical application of SPOINS for MRI imaging. 

Minor comments:

Figures 6 and 7 need to be moved to the results section.

Line No. 26: change to “SPIONS are not affecting the cell viability”

Line 39: change to “classified into two categories”

Line 42: change “biomedical applications such as”

Line 50: consider revising this sentence, it is not clear

Line 265-279: This section including Fig 6, can be moved to results section, as results and discussion sections are separated in this manuscript.

Author Response

In this manuscript by Han and Co-authors presented their results on the newly synthesized dextran coated SPOINS with carboxymethyl group, chelated with europium complexes.   These newly synthesized SPOINS were thoroughly characterized and evaluated for their biocompatibility using the HEK293T and HepG2 cells.  Biocompatibility studies included multiple assays which includes cytotoxicity by WST and apoptosis using Annexin V.   Overall the manuscript adds to the biomedical application of SPOINS for MRI imaging. 

Minor comments:

1. Figures 6 and 7 need to be moved to the results section.

Response 1: Thanks for the comments. The Figure 6, 7 and some description were moved to “results” section. 

2. Line No. 26: change to “SPIONS are not affecting the cell viability”

Response 2: Thanks for the comments. The sentence was changed.

3. Line 39: change to “classified into two categories”

Response 3: Thanks for the comments. The sentence was changed.

4. Line 42: change “biomedical applications such as”

Response 4: Thanks for the comments. The sentence was changed.

5. Line 50: consider revising this sentence, it is not clear

Response 5: Thanks for the comments., The sentence was revised as following; “the lanthanide complexes have been intensively investigated in nanomedicine “

6. Line 265-279: This section including Fig 6, can be moved to results section, as results and discussion sections are separated in this manuscript.

Response 6: Thanks for the comments. The Figure 6, 7 and some description were moved to “results” section. 

Reviewer 3 Report

The manuscript is well prepared. The article is devoted to multi-functional nanoparticles and will be of interest to many readers. The manuscript may be accepted after correction of minor inaccuracies.

Some corrections

91 mmol. From the stock solution, 2 mL Eu3+, 6 mL TFAAN, 6 mL P(Oct)3 and 100 uL NH4OH were

What is concentration of NH4OH? 100 uL M??? NH4OH

Line 97-98

Scheme 1. Schematic illustration of (a) formation of europium complexes with 2-(4,4,4-Trifluoroacetoacetyl)naphthalene (TFAAN)

The formula 2- (4,4,4-Trifluoroacetoacetyl)naphthalene (TFAAN) is described in the figure caption, but another beta-diketone (2-(4,4,4-Trifluoroacetoacetyl)thiophene) is drawn.

Unnecessary hydrogen atom at phosphorus is in Figure 1C

196 and manipulate the stability of colloidal dispersion. The ζ-potentials of CMD-SPIONs is 9.7 mV and  

197 Eu(TFAAN)3(P(Oct)3)3 chelated on CMD-SPIONs is changed to -17 mV.

The ζ-potentials of CMD-SPIONs have to be negative, but Eu(TFAAN)3(P(Oct)3)3 chelated on CMD-SPIONs can have positive charge associated with trioctylphosphine in the complex.

202 bond. The stretching vibration bands of P=O (1,061 cm−1) bond are appeared in Eu(TFAAN)3(P(Oct)3)3

203 and Eu(TFAAN)3(P(Oct)3)3 chelated on CMD-SPIONs. These results imply that Eu(TFAAN)3(P(Oct)3)3

Group P=O is absent in the structure of the complex.

215 of epichlorohydrin reacts with the -OH group to form a liberated chlorohydrin on the dextran.

216 Afterward, a dechlorination occurs between -OH group on the dextran and the liberated  chlorohydrin

217 to form the corresponding crosslinked arrangement with a new -OH group at the  terminal.

Have to be

of epichlorohydrin alkylates OH groups and its epoxy group interacts with other OH groups of the dextran to form the corresponding inter- and intramolecular crosslinks.

Author Response

The manuscript is well prepared. The article is devoted to multi-functional nanoparticles and will be of interest to many readers. The manuscript may be accepted after correction of minor inaccuracies.

Some corrections

1. 91 mmol. From the stock solution, 2 mL Eu3+, 6 mL TFAAN, 6 mL P(Oct)3 and 100 uL NH4OH were

What is concentration of NH4OH? 100 uL M??? NH4OH

Response 1: Thanks for the comments. Remarked the concentration in sentence “(28% in water)”

2. Line 97-98

Scheme 1. Schematic illustration of (a) formation of europium complexes with 2-(4,4,4-Trifluoroacetoacetyl)naphthalene (TFAAN)

The formula 2- (4,4,4-Trifluoroacetoacetyl)naphthalene (TFAAN) is described in the figure caption, but another beta-diketone (2-(4,4,4-Trifluoroacetoacetyl)thiophene) is drawn.

Unnecessary hydrogen atom at phosphorus is in Figure 1C

Response 2: Thanks a lot for the comments. It was big mistake. The scheme was modified and Unnecessary hydrogen atom at phosphorus in Figure 1C was deleted.

3. 196 and manipulate the stability of colloidal dispersion. The ζ-potentials of CMD-SPIONs is 9.7 mV and  

197 Eu(TFAAN)3(P(Oct)3)3 chelated on CMD-SPIONs is changed to -17 mV.

The ζ-potentials of CMD-SPIONs have to be negative, but Eu(TFAAN)3(P(Oct)3)3 chelated on CMD-SPIONs can have positive charge associated with trioctylphosphine in the complex.

Response 3: Thanks a lot for the comments. It was mistake. Data was changed. The sentence was modified to “The ζ-potentials of CMD-SPIONs is -17 mV and Eu(TFAAN)3(P(Oct)3)3 chelated on CMD-SPIONs is changed to 9.7 mV.” Also, remarks in Figure 4 (a) were properly corrected.

4. 202 bond. The stretching vibration bands of P=O (1,061 cm−1) bond are appeared in Eu(TFAAN)3(P(Oct)3)3

203 and Eu(TFAAN)3(P(Oct)3)3 chelated on CMD-SPIONs. These results imply that Eu(TFAAN)3(P(Oct)3)3

Group P=O is absent in the structure of the complex.

Response 4: Thanks for the comments. The concentration of europium complex was too low than CMD-SPIONs. We increased the amount of europium complex and measured the FTIR analysis again. As you indicated that P=O peak clearly appear in new measurement.

5. 215 of epichlorohydrin reacts with the -OH group to form a liberated chlorohydrin on the dextran.

216 Afterward, a dechlorination occurs between -OH group on the dextran and the liberated chlorohydrin

217 to form the corresponding crosslinked arrangement with a new -OH group at the  terminal.

Have to be of epichlorohydrin alkylates OH groups and its epoxy group interacts with other OH groups of the dextran to form the corresponding inter- and intramolecular crosslinks.

Response 5: Thanks a lot for the comments. The sentence was replaced based on the comments.